# Biofilm Inhibitory Activity of Actinomycete-Synthesized AgNPs with Low Cytotoxic Effect: Experimental and In Silico Study

**DOI:** 10.3390/microorganisms11010102

**Published:** 2022-12-30

**Authors:** Sabah A. AboElmaaty, Ali A. Shati, Mohammad Y. Alfaifi, Serag Eldin I. Elbehairi, Norhan S. Sheraba, Mervat G. Hassan, Mona Shaban E. M. Badawy, Ahmed Ghareeb, Ahmed A. Hamed, Ebtsam Z. Gabr

**Affiliations:** 1Botany and Microbiology Department, Faculty of Science, Benha University, Benha 13511, Egypt; 2Department of Biology, Faculty of Science, King Khalid University, Abha 9004, Saudi Arabia; 3Cell Culture Lab, Egyptian Organization for Biological Products and Vaccines (VACSERA Holding Company), Giza 12511, Egypt; 4VACSERA, The Holding Company for Biological Products and Vaccines, Giza 12511, Egypt; 5Department of Microbiology and Immunology, Faculty of Pharmacy (Girls), Al-Azhar University, Cairo 11884, Egypt; 6Botany and Microbiology Department, Faculty of Science, Cairo University, Giza 12613, Egypt; 7Microbial Chemistry Department, National Research Center, 33 El-Buhouth Street, Giza 12622, Egypt

**Keywords:** biogenic AgNPs, *streptomyces*, antibifilm

## Abstract

The emergence of resistance by biofilm-forming bacteria has reached alarming and dangerous levels that threaten human civilization. The current study sought to investigate the antibiofilm potential of green-synthesized silver nanoparticles, mediated by a new *Streptomyces* strain. Zeta potential, transmission electron microscopy (TEM), and UV-Vis spectroscopy were used to analyze the biosynthesized AgNPs. Results revealed that silver nanoparticles had a size of (5.55 and 45.00 nm) nm and a spherical shape, with surface plasmon resonance (SPR) absorption at 400–460 nm in the UV-vis spectra establishing the formation of *Streptomyces*-Ag-NPs. The biosynthesized AgNPs showed a pronounced antibacterial efficacy against *Escherichia coli*, *Pseudomonas aeruginosa*, *Bacillus subtilis*, and *Staphylococcus aureus.* Moreover, the obtained *Streptomyces*-AgNPs exerted biofilm inhibition activity against nosocomial hospital-resistant bacteria, including *Bacillus subtilis*, *Staphylococcus aureus*, and *Escherichia coli*. The mechanism of biogenic AgNPs antibacterial action was visualized using TEM, which indicated the AgNPs accumulation and disruption of bacterial cell membrane function. Additionally, a molecular docking study was conducted to evaluate the binding mode of AgNPs with an Escherichia coli outer membrane. Furthermore, the cytotoxic profile of the AgNPs was evaluated toward three cell lines (MCF-7, HepG2 & HCT 116), and the low cytotoxic effects of the obtained nanoparticles indicated their possible medical application with low risks to human health.

## 1. Introduction

Antimicrobial resistance (AMR) is one of the greatest public health challenges that currently threaten the successful prevention and treatment of a rising variety of bacterial, parasitic, viral, and fungal infections [1]. Nosocomial pathogenic bacteria became a significant obstacle to medical remediation after the emergence of antibiotic-resistant bacteria, with new resistance mechanisms emerging as a result of the abuse of antibiotics. [2,3,4]. One of these mechanisms is the development of bacterial biofilms, in which bacteria employ a special form of communication known as quorum sensing (QS). This enables other bacteria to react and cooperate to create bacterial biofilm aggregations [5,6]. This biofilm is made up of different macromolecules known as extracellular polymeric substances (EPS), types of tertiary biofilm structure that allow surface adherence and cell-to-cell interaction [6,7]. This matrix protects dangerous bacteria against potential antibiotic damage and attacks from the human immune system, making it one of the most serious challenges to medication and infectious disease prevention [7,8,9].

Despite many attempts that have been conducted to eradicate biofilm formation, there is still a definite need to discover an efficient agent to control biofilm formation by bacteria [10]. In hospitals and other healthcare facilities, the rate of nosocomial infections caused by antibiotic-resistant pathogens has increased [11]. The three most prevalent bacterial pathogens that cause nosocomial infections are *E. coli*, *Klebsiella* sp., and *S. aureus* [12,13,14,15].

Nanostructured silver is one of the noblest metals that has recently been used in several applications, starting from agriculture and ranging to medical use [16]. It has been estimated that the global world demand for nanomaterials ranges from 0.3 million tons to 1.6 million tons, including the Asian region, which accounts for approx. “(34%)”, followed by North America (about 31%) and Europe (about 30%), respectively [16,17,18]. Silver nanoparticles (AgNPs) are structures with unique physical, chemical, and biological features of less than 100 nm [19,20,21]. AgNPs have been recently proven to be very good candidates as antimicrobial agent against some serious superbugs including *E. coli*, *S. typhi*, *S. epidermidis*, and *S. aureus*.

In this work, we examined a new actinomycete strain for its ability to perform bio-synthesis of an eco-friendly silver nanoparticle (AgNPs) from a cell supernatant of *Streptomyces* sp. ASM23 (CS-AgNPs) and cell filtrate (CF-AgNPs). The biosynthesized AgNPs exhibited a remarkable biofilm inhibitory activity toward some tested nosocomial-resistant bacteria, including *B. subtilis*, *S. aureus*, and *E. coli*. Furthermore, we reported the in vitro cytotoxicity of the *streptomyces*-biosynthesized AgNPs against three cell lines (MCF-7, HepG2 & HCT 116), and the obtained results showed that the cell filtrate AgNPs (CF-AgNPs) have low cytotoxic activity. This suggests the possible use of biosynthesized nanometal in the medical field. Moreover, we studied the mechanism of inhibitory activity in silico aside from microscopic visualization.

## 2. Materials and Methods

### 2.1. Chemicals

All reagents were of analytical grade and purchased from Sigma–Aldrich (Burlington, MA, USA) and Himedia (Mumbai, India).

### 2.2. Microbial Strains and Cell Cultures

*Bacillus subtilis* ATCC6633, *Escherichia coli* ATCC25955, *Staphylococcus aureus* ATCC 33591, *Pseudomonas aeruginosa* ATCC10145, *Candida albicans* ATCC10231, and *Aspergillus niger* ATCC 6275 were obtained from the Microbiology and Immunology Department, Faculty of Medicine, Al-Azhar University, Egypt. Human breast adenocarcinoma (MCF-7), hepatocellular carcinoma (HepG-2), and colorectal adenocarcinoma (HCT 116) cell lines were obtained from the American Type Culture Collection (ATCC), USA. Cells were cultured in RPMI-1640 supplemented with streptomycin (100 μg/mL), and then with penicillin (100 units/mL) and heat-inactivated fetal bovine serum (10% *v*/*v*) in a humidified, 5% (*v*/*v*) CO_2_ atmosphere at 37 °C.

### 2.3. Sample Collection and Isolation of the Bacterial Strain

Soil samples were collected from the Dakahlia Governorate, Egypt. Each sample was placed in clean, sterile bottles stored at 4 °C. Via intensive vortexing, the soil samples were suspended in sterile distilled water, and serial dilutions were made in sterile distilled water [18]. A volume of 0.1 mL of appropriate dilution was added to the Petri plate containing a starch nitrate agar media. Incubation was performed at 30 °C, and pH 7 was observed for seven days until the growth of actinomycetes [22,23]. A well-isolated formed colony was picked and purified by streaking on a starch nitrate agar medium.

### 2.4. Phenotypic Characterization of the Isolated Strain

Cultural characteristics studies of the ASM23 isolate have been studied according to the guidelines set out in the International *Streptomyces* Project. After its growth on various (ISP) media, the isolate was observed for its morphological features, pigmentation, and aerial mycelium. The microscopic examination was carried out to visualize the spore chain morphology of the bacterial isolate on starch nitrate agar plates [24,25]. The electron micrograph was captured by a Transmission Electron Microscope (TEM) at the National Research Centre (NRC), Egypt. Moreover, the utilization of various carbon sources was also studied.

### 2.5. Molecular Identification of Isolated Strain

The genetic identification of the isolated actinomycete strain was carried out through 16S rRNA gene sequencing. DNA extraction was performed according to the manufacturer′s instructions using the DNeasy Blood & Tissue Kit (QIAGEN). Two universal primers (27F-5′-AGAGTTTGATCCTGGCTCAG-3′ and 1492R 5′GGTTACCTTGTTACGACTT-3′) were used to conduct 50 µL PCR amplification reactions. The purified PCR products were then sent to the Macrogen Company, South Korea, for sequencing [26]. The sequence was then investigated for homology and similarity using online NCBI BLAST search tools to compare similar known sequences in the NCBI database. The phylogenetic trees have been built using MEGA-X software.

### 2.6. Actinomycetes Cell-Free Extract Preparation

The strain *Streptomyces* sp. ASM23 was cultivated in yeast malt extract broth with the following composition g/L: yeast extract, 4.0; malt extract, 10.0; dextrose, 4.0. The broth had a pH of 7.2, and the strain was incubated at 32 °C in a rotary shaker at 150 rpm for six days. After incubation, the bacterial cell pellets were separated from the culture supernatant by centrifuging the whole content at 10,000 rpm for 15 min. To prepare the cell filtrate, 1 g of the collected pellets was re-suspended in 50 mL sterile distilled water, incubated for 48 h at 28 °C under shaking conditions (150 rpm), and centrifuged at 10,000 rpm for 15 min [24].

### 2.7. Green Synthesis of Silver Nanoparticles Using Actinomycetes Extracts

AgNPs biosynthesis was carried out by mixing an equal volume of bacterial supernatants and cell filtrate at a final concentration (1.0 mM). The mixture was incubated in a dark rotary shaker (150 rpm) for 3 days at 32 °C. The control experiment was conducted to determine the role of bacteria in the biosynthesis of AgNPs by mixing uninoculated culture media with the solution of AgNO_3_. After incubation with actinomycetes supernatant and cell filtrate, the reduction of silver ions in the incubated solution was followed by a color change from colorless to yellowish-brown. The UV–Vis spectra of the incubated solution at different time intervals were measured using (SPECTROstar Nano absorbance plate reader—BMG LABTECH, Offenburg, Germany) with an increment of 2 nm and with a slot width of 5 nm in the 200 to 900 nm range [25,26,27,28].

### 2.8. Transmission Electron Microscopy (TEM)

Transmission electron microscopy (TEM) was used to measure the size and morphology of the synthesized AgNPs. A volume of 2 μL of AgNPs solution was mounted on carbon-coated copper grids to prepare the sample. After adding the sample, the thin film formed was air-dried and observed using a Philips 10 Technai with an accelerating voltage of about 180 keV and with a wavelength (λ) of 0.0251 Å at the National Research Center, Egypt.

### 2.9. Studies of X-ray Diffraction (XRD)

The X-ray diffraction (XRD) pattern of the generated green AgNPs was measured using a PAN analytical X’pert PRO X-ray diffractometer (Philips, Eindhoven, The Netherlands) at the National Research Center, Egypt, with Cu Ka1 radiation. It was determined that there was an operating voltage and tube current of around 40 kV and 30 mA, respectively. After the material was drop-coated onto a glass substrate, the X-ray diffraction patterns were collected at 2θ from 10° to 80° at a scanning speed of 0.02°/min.

### 2.10. Zeta Potential

Zeta potential measurements are done using a high-performance two-angle Zetasizer Nano ZS device at the National Research Center, Egypt.

### 2.11. Determination of the Antimicrobial Activity of (AgNPs)

The antimicrobial activity of the biosynthesized AgNPs was measured using the agar well diffusion method. To measure the antimicrobial activity, a set of pathogenic test bacteria comprising Gram-positive bacteria (*Staphylococcus aureus* and *Bacillus subtilis*), Gram-negative bacteria (*Pseudomonas aeruginosa* and *Escherichia coli*), yeast (*Candida albicans*), and the fungus (*Aspergillus niger*) were selected. Both bacterial and yeast microbes were grown on a nutrient agar medium: this was composed of 10 g L^−1^ peptone, 3 g L^−1^ beef extract, and 20 g L^−1^ agar. The pH was adjusted to 7.2. On the other hand, the fungal strain was cultivated on a potato dextrose agar medium (PDA) of the following ingredients: 4 g L^−1^ potato extract, 20 g L^−1^ dextrose, and 20 g L^−1^ agar (pH 6). The culture of each microorganism was diluted with sterile distilled water to concentrations ranging from 10^7^ to 10^8^ CFU/mL. After preparing the inoculated agar plates, wells of 5 mm diameter were punched into the agar medium and filled with 50 μL of conc. and 50 µg/mL of the biosynthesized AgNPs and allowed to diffuse at 5 °C for 2 h. The plates were incubated for 24 hrs at 37 °C for bacteria and 48h at 30 °C for fungi and yeast. The inhibition zone diameter was measured after the incubation period [29]. Ciprofloxacin (5 µg/disc) and nystatin (10 µg/disc) were used as positive controls.

### 2.12. Monitoring Microbial Growth Curve Dynamics for E. coli in the Presence of Silver Nanoparticles Solutions

The most successful treatment was subjected to further investigations of the effect of different AgNPs concentrations along the *E. coli* growth curve. The antibacterial effectiveness of different concentrations of filtrate AgNPs’ solution (0, 20, 40, 60 and 80, 100 µg/mL) was tested to examine its activity on the bacterial growth curve for *E. coli* bacterial colonies. These were inoculated with nutrient broth media till reaching optical culture density (OD600) 0.25–0.35. Different concentrations of AgNPs’ solution (0, 20, 40, 60, 80, and 100 µg/mL) were added to the inoculated cultures. The cultures were incubated at 37 °C at 180 rpm. Bacterial densities were followed by determining the optical density (OD) at 600 nm after 0, 24, 48, 72, and 96 hrs [30].

### 2.13. In Vitro Antibiofilm Activity

The formed AgNPs’ inhibitory biofilm activity was evaluated by applying a microtiter plate (MTP) assay to four resistant bacteria obtained from Al-Azhar university hospitals in Egypt (*Escherichia coli*, *Pseudomonas aeruginosa*, *Staphylococcus aureus*, and *Bacillus subtilis*). Briefly, each well was filled with LB broth (180 μL). It was then inoculated with pathogenic bacterial suspension (10 μL), 10 µL of (50 μg/mL) AgNPs was added along with the control (without the test sample) and the solution was incubated statically at 37 °C for 24 h. After the incubation period, the floating bacterial content was carefully removed and well washed with 200 µL of phosphate buffer saline (PBS) pH 7.2, left to dry, and stained with 200 µL/well of crystal violet solution (0.1%, *w*/*v*) for 10 min. The plate wells were washed with 200 µL/well-distilled water three times and kept for drying. To solubilize the dye, 150 µL/well of 95% ethanol was added, and the optical density at 570 nm was measured by (SPECTROstar Nano absorbance plate reader (BMG LABTECH)). All isolates were duplicated at least four times [31].

### 2.14. In Vitro Cytotoxic Activity

The cytotoxicity of the biosynthesized AgNPs from *Streptomyces* sp. ASM23 culture supernatant and cell filtrate was assessed against human tumor cells (MCF-7, HepG-2 & HCT 116) using the Sulphorhodamine B assay (SRB). Additionally, 90% confluence growing cells were trypsinized and cultured for 24 h before treatment with the biosynthesized AgNPs in a 96-well tissue culture plate (3000 cells/well). Cells were exposed to the six different concentrations of each compound (0.01, 0.1, 1, 10, and 1000 µg/mL); untreated cells (control) were added. The cells were incubated with the concentrations for 72 h and subsequently fixed with TCA (10% *w*/*v*) for 1 h at 4 °C. After washing, cells were stained with 0.4% (*w*/*v*) SRB solution for 10 min in a dark place. The excess stain was washed with 1% (*v*/*v*) glacial acetic acid. The SRB-stained cells were dissolved with Tris-HCl after drying overnight, and the color intensity in the microplate reader was measured at 540 nm. The relation between percentages of cell line viability and concentrations of the compounds has been analyzed to determine the IC_50_ (drug dose that decreases survival to 50%) using SigmaPlot 12.0 software.

### 2.15. Molecular Docking

The outer membrane proteins (OMPs) of Gram-negative bacteria play an important role in mediating antibacterial resistance and bacterial virulence, thus affecting the pathogenic ability of the bacteria. The outer membrane proteins and OMPX (1QJ8) crystal structure were obtained from the PDB database. The three-dimensional structure of this protein was taken as the target for molecular docking, with Ag using the PatchDock server [32]. The SDF file of silver was downloaded from the PubChem database and converted into a PDB file using online SMILES translator (https://www.cactus.nci.nih.gov/translate/, accessed on 12 October 2021). The PDB files of the respective protein and silver were uploaded under the receptor–ligand interaction mode, with default values and an RMSD cutoff value of <4.0 Å. Based on the scoring, the top-ranked solution was selected for interpretation. The amino acid residues of the proteins that are within 4 Å were analyzed for their molecular interaction with silver.

### 2.16. Statistical Analysis

Data were analyzed using the one-way analysis for variance (ANOVA) method and carried out by the SPSS program based on the mean of the triplicate results.

## 3. Results

### 3.1. Collection and Isolation of Bacterial Isolate

A single colony of strain ASM23 was isolated using serial dilution techniques from the soil collected from Mansoura. The isolate was coded, retained at 4 °C on starch nitrate agar slants, and deposited at the Culture Collection Center, Department of Microbial Chemistry, National Research Center, Egypt.

### 3.2. Phenotypic Studies of Strain ASM23

Morphological properties of the isolated strain ASM23 on different ISP media (ISP2-ISP5) showed that the strain has very close characteristics to members of the genus *Streptomyces* [24,25]. The aerial mycelium of the strain ASM23 is mainly whitish-grey on different ISP media, with no diffusible pigments (Figure 1a). Microscopic examination revealed that the spore of the ASM23 is short with a smooth spore surface (Figure 1b). On the other hand, the strain utilizes D-glucose, sucrose, D-Fructose, and l-arabinose as carbon sources.

### 3.3. Molecular Identification of Selected Actinomycetes

Sequencing of 16S rDNA has been carried out to identify the ASM23 isolate. The DNA was isolated, amplified using PCR, sequenced, and aligned using the NCBI Online BLAST tool to examine similarity scores with other identified strains. The results confirmed a high similarity rating of 99.72% homology of isolate ASM23 with *Streptomyces* sp. The partial 16S rDNA sequence of the *Streptomyces* isolate was deposited in the GenBank with the MN689775 accession number under the title *Streptomyces* sp. ASM23. The phylogenetic trees have been built by MEGA-X and a maximum likelihood method (Figure 2) [33].

### 3.4. Green Biosynthesis of AgNPs

The green biosynthesis of silver nanoparticles AgNPs was performed using *Streptomyces* sp. ASM23 supernatant (CS) and cell filtrate (CF). The synthesis was performed under dark conditions at 32 °C. A solution of a brownish color forms after the addition of AgNO3 aqueous solution (1 mM) to the *Streptomyces* sp. ASM23 supernatant and cell filtrate, implying metal ions reduction and the formation of AgNPs (Figure 3b,c). AgNPs have light yellow to brown colors due to the vibration of surface plasm in the particles confirmed by spectra for UV-Vis absorption (Figure 3a). The biosynthesized AgNPs in the presence of actinomycete cell filtrate exhibited a characteristic surface plasmon band (SPR) at a wavelength ranging from 400 to 460 nm, indicating the formation of AgNPs at different time intervals. These findings follow early reports by Składanowski et al. [34] and Al-Dhabi et al. [35].

### 3.5. Characterization of Silver Nanoparticles

Dynamic light scattering (DLS) and zeta potential measurements were used to investigate the biosynthesized AgNPs’ surface and colloidal stability. The results indicated that the particle size (i.e., hydrodynamic, HD diameter) increased when adding the cell supernatant and cell filtrate of *Streptomyces* sp. ASM23 to the aqueous solution of AgNO_3_. This results from an Ag^+^ ion being reduced in Ag nuclei and forming large particles. Moreover, through *Streptomyces* sp. ASM23, the zeta potential for the AgNPs formed by the aqueous solution of the AgNO_3_ samples was −27.6 and −30.5 mV, respectively. The results showed that the prepared AgNPs tend to attract particles electrostatically in the solution. The particle size and shape have been demonstrated via transmission electron microscope (TEM) measurements for two prepared samples (CS-AgNPs and CF-AgNPs), as shown in (Figure 3e,f). The average particle size for silver nanoparticles biosynthesized by the *Streptomyces* sp. The particle sizes ASM23 cell-free filtrate and supernatant range from about ~5.55 to 35 nm, with spherical and cubic shapes (Figure 3d–f). The X-ray diffraction (XRD) pattern of silver nanoparticles (NPs) was measured and presented in Figure 3g. Peaks of XRD are assigned to diffraction from the (111), (200), and (220) planes of silver.

### 3.6. Antimicrobial Activity of Biosynthesized AgNPs

The antimicrobial activity of the synthesized silver nanoparticles has been measured toward different human pathogens using the agar well diffusion method. In general, the AgNPs, prepared by the cell filtrate of *Streptomyces* sp. ASM23, showed more antimicrobial activity than the supernatant solutions. The results showed that *E. coli* and *S. aureus* were the most susceptible bacteria to the antibacterial effects of AgNPs, prepared by *Streptomyces* sp. ASM23 cell filtrate (CF-AgNPs), with inhibition zone diameters of 19.00 and 16.00 mm, respectively. On the other hand, the formed CF-AgNPs displayed moderate antimicrobial activity toward *P. aeruginosa* and *B. subtilis*. Additionally, the AgNPs of both filtrate and supernatant showed good antibacterial activity against *S. aureus* while exhibiting low activity against *C. albicans* and no response was detected against *A. niger* (Table 1).

### 3.7. Effect of Different Concentrations of AgNPs on E. coli Growth Curve Dynamics

The effect of CF-AgNPs, biosynthesized by the filtrate solution on the growth curve dynamics of *E. coli*, has been investigated. The bacterial growth curve slope dwindled perpetually with elevating nanoparticle concentration. The highest inhibition activity has been detected at an 80 µg/mL concentration, which started with a dwarfed long period. At 80 mg/mL of AgNPs, the most marked effect is easily visualized in the late logarithmic phase and during the bacterial stationary growth phase (Figure 4).

### 3.8. Antibacterial Mechanism of Action

The antibacterial mechanism of the AgNPs has been visualized using transmission electron microscopy (TEM). The *S. aureus* was treated with CF-AgNPs at a final concentration, of 50 µg mL^−1^, and incubated at 37 °C overnight. The TEM micrograph displayed the accumulation of CF-AgNPs on the *S. aureus* cell membrane, broken membranes of the treated *S. aureus*, and the appearance of CF-AgNPs inside the bacterial cells (Figure 5a,b).

### 3.9. Antibiofilm Activity of Biosynthesized AgNPs

The AgNPs, synthesized by the *Streptomyces* sp. ASM23 cell filtrate (CF-AgNPs), displayed a potent biofilm inhibition activity against *E. coli* and *B. subtilis*, with biofilm inhibitory ratios of 74.20 and 70.52%. At the same time, the AgNPs synthesized by *Streptomyces* sp. ASM23 cell filtrate (CF-AgNPs) only displayed a moderate antibiofilm activity toward *S. aureus*, up to 50.30%. On the other hand, *P. aeruginosa* displayed an inadequate response, with a biofilm formation inhibition rate up to 20.00% (Figure 6). Statistical analysis of the biofilm data, based on mean of triplicate and standard deviations (SD), showed that the p value is lower than 0.05. This indictaed that the result is statistically significant.

### 3.10. In Vitro Cytotoxic Activity

The sulforhodamine B (SRB) assay was used to assess the cytotoxicity of the green CF-AgNPs and CS-AgNPs towards MCF-7, HepG2, and HCT 116 tumor cell lines over a concentration range from 0.01 to 1000 μg/mL (Table 2 and Figure 7 and Figure 8). Tested compounds showed a variety of cytotoxicity profiles against solid tumor cells. Biosynthesized AgNPs from the cell supernatant of *Streptomyces* spp ASM23 showed a potent cytotoxic profile in all tested cell lines (MCF-7, HepG2 & HCT 116), with IC_50_s of 27.8, 18, and 73.2, respectively, more than biosynthesized AgNPs from the cell filtrate of *Streptomyces* spp ASM23. The toxicity of biosynthesized AgNPs from the cell supernatant of *Streptomyces* spp ASM23 on HCT 116 was weaker compared to MCF-7 and HepG2 cells, whereas biosynthesized AgNPs from cell filtrate of *Streptomyces* spp ASM23 have shown promising toxicity against MCF-7 cells with IC_50_ 65.1, relevant to HepG2 and HCT 116 cells with IC_50_s 82.9 and 81.9, respectively, shown in (Table 2, Figure 7 and Figure 8).

### 3.11. Molecular Docking Determine the AgNP’s Binding Capacity to E. coli

The antibacterial effect was visualized and confirmed using a TEM micrograph. This showed the accumulation of formed Ag nanoparticles on bacterial cell walls that consist of peptidoglycan; these work in turn as an essential protective barrier for bacteria and encapsulate the cytoplasmic membrane of both Gram-negative and Gram-positive bacterial cells. To assess and predict the antibacterial activity of silver nanoparticles, molecular docking was performed using the PatchDock server for the three-dimensional structures of *E. coli* outer membrane proteins, the OMPX (1QJ8) crystal structure. The amino acid residues that interact with Ag, along with the type of interactions, are presented in Figure 9. Ag interacted with OMPX (1QJ8) through Phe148, which may lead to disruption of the *E. coli* outer membrane.

## 4. Discussion

Despite the pioneer achievements in organic and synthetic chemistry to develop effective antimicrobial agents, new candidates with novel action mechanisms are still urgently needed to counter the ongoing increase in antibiotic resistance due to biofilm formation. AgNPs have unique physical and optical properties that make them suitable for a wide range of potential applications such as antimicrobial agents, coating of biomedical devices, and drug nano-delivery approaches [19]. Recently, AgNPs’ antibacterial effects have been the subject of current controversy for the inhibition of most common nosocomial pathogens that cause various clinical diseases such as *E. coli*, *K. pneumonia* and *S. aureus* [36].

The in vitro microbicidal activity of the AgNPs was previously tested against diverse pathogenic fungi such as *F. solani*, *A. niger*, *A. parasiticus*, and *C. albicans*, and different pathogenic bacteria such as *B. megaterium*, *Bacillus subtilis*, *Proteus vulgaris* and, *Escherichia coli*. Chudasama et al. and Ramgopal et al. [6,37] have stated that AgNPs displayed inhibitory activity toward *S. aureus* (Gram-positive) and *E. coli* (Gram-negative). Prakash et al. [38] have also highlighted a similar result for the antibacterial effects of the AgNPs on Gram-negative bacteria (*E. coli*) and Gram-positive (*S. pyogene*). Conversely, Dipak and Sankar et al. [39] have shown the highest and the lowest zone of inhibition formation against *E. coli* and *Staphylococcus* sp., respectively. The obtained results showed that AgNPs, prepared by *Streptomyces* sp. ASM23 cell filtrate, have antibacterial activity against *E. coli* and *S. aureus*, while it showed moderate antibacterial activity against *B. subtilis*. Several studies have reported the possible antimicrobial mechanism of silver nanoparticles and reported the AgNPs’ primary mechanism of action.

Therefore these particles can penetrate the cell membrane, or attach themselves according to their size to the bacterial surface [7]. Moreover, the bactericidal efficacy is further increased by reducing the size of the particle [40]. There have been many arguments to clarify why microbial growth inhibition by AgNPs should occur; the most convincing is free radical formation, a process which was supported in the spectrum of AgNPs by the peak appearance at 336.33 in the electron spin resonance (ESR) spectrum of AgNPs [41]. The generation of free radicals primarily disrupts the microbial membrane lipids structure, dissociate it, and hinders microbial growth [42]. In one study, the release of silver cations from AgNPs was attributed to the high antibacterial activity [43]. These Ag^+^ ions are charged positively and far less than neutral AgNPs. They can easily interact with the S or P and N electron-rich biomolecules in the bacterial cell walls. Some researchers have hypothesized that a key to microorganism growth inhibition is the interaction between the positive AgNPs’ charge and the negative charge on the cell membrane [44,45,46]. The nanoparticles form pits within the microbial cell wall; they then accumulate and eventually permeate into the bacterial cells [41,47]. In the presented work, analysis by TEM micrograph proved the accumulation of AgNPs, both on the bacterial cell membrane and inside the bacterial cells. Some reports declared that the AgNPs adhere primarily to the bacterial surface and alter membrane properties, but that they may cause DNA damage inside the bacterial cell [48]. McQuillan and colleagues stated that cell membrane dissolution is the primary mechanism for the action of silver nanoparticles [49]. Several reports suggested that silver nanoparticles’ mechanism of inhibition started through the binding and interacting of silver with thiol-containing proteins in the cell wall and influencing their functions [50].

Computational study, carried out through omlecular docking, was performed using a PatchDock server for the three-dimensional structures of *E. coli* outer membrane proteins with an OMPX (1QJ8) crystal structure. The results showed that the biosynthesized AgNPs bind to 1QJ8 through Phe146, an occurrence which may be the cause of disrupting the *E. coli* outer membrane. Additionally, the antibiofilm activity of the formed silver nanoparticles against a set of hospital nosocomial bacterial pathogens such as *B. subtilis* and *E. coli* indicated its possible use in the medical field. Moreover, the low cytotoxic effect of AgNPs, synthesized via a filtrate of *Streptomyces* sp. ASM23 against all tested cell lines (MCF-7, HepG2 & HCT 116), makes them promising and possible candidates for application in the biomedical field. Additionally, they have a strong safety profile.

## 5. Conclusions

The spread of antimicrobial resistance has made it imperative to look for an alternative source for an adequate drug to combat multi-drug-resistant bacteria. The obtained results concluded that actinomycetes could be an effective, cheap, environmentally friendly, and secure strategy for the green biosynthesis of nanoparticles. Additionally, silver nanoparticles showed pronounced antibiofilm activity against a set of hospital nosocomial tests pathogenic such as *B. subtilis* and *E. coli*, indicating their possible applicability in the medical field. Moreover, AgNPs synthesized via filtrates of *Streptomyces* sp. ASM23 showed a low cytotoxic profile in all tested cell lines (MCF-7, HepG2 & HCT 116), reflecting their possible use in the biomedical field and their high safety profile. The obtained results highlighted the possible use of green AgNPs as an anti-biofilm agent with an efficient effect on hospital nosocomial bacterial pathogens.

## Figures and Tables

**Figure 1 microorganisms-11-00102-f001:**
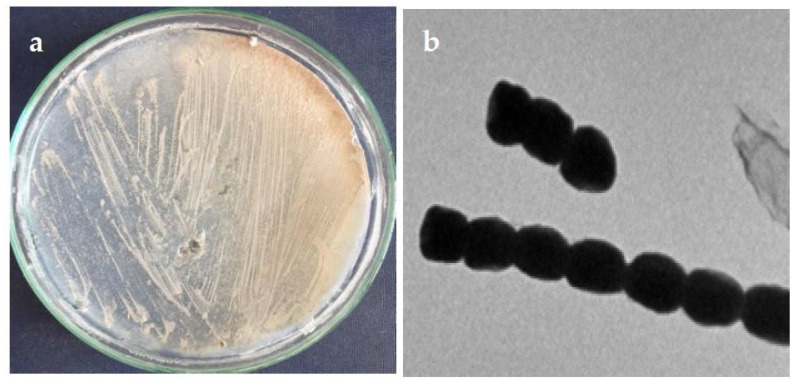
(**a**) Aerial and substrate mycelium morphology; (**b**) spore chain morphology under transmission electron micrography.

**Figure 2 microorganisms-11-00102-f002:**
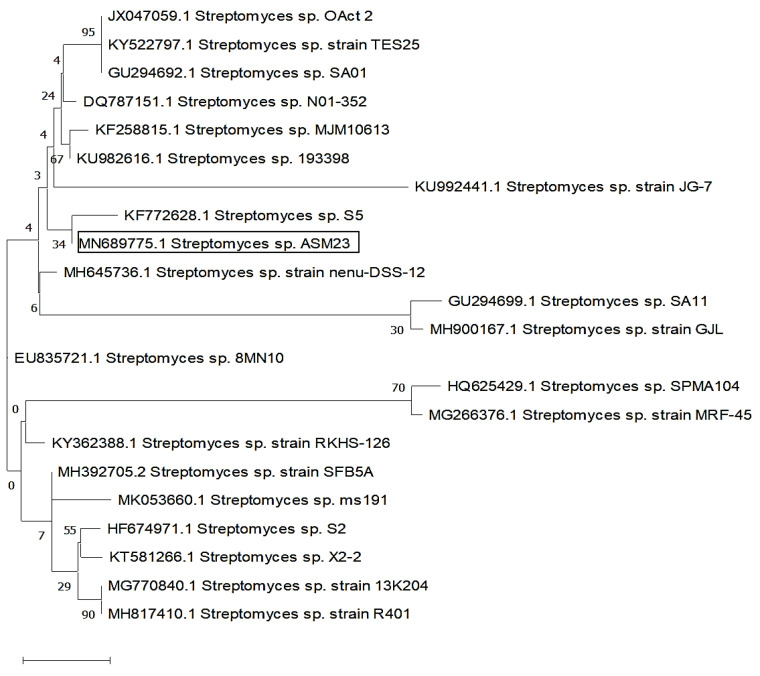
Evolutionary relationships of taxa related to *Streptomyces* sp. ASM23 is based on a 16S rDNA sequence using the maximum likelihood method.

**Figure 3 microorganisms-11-00102-f003:**
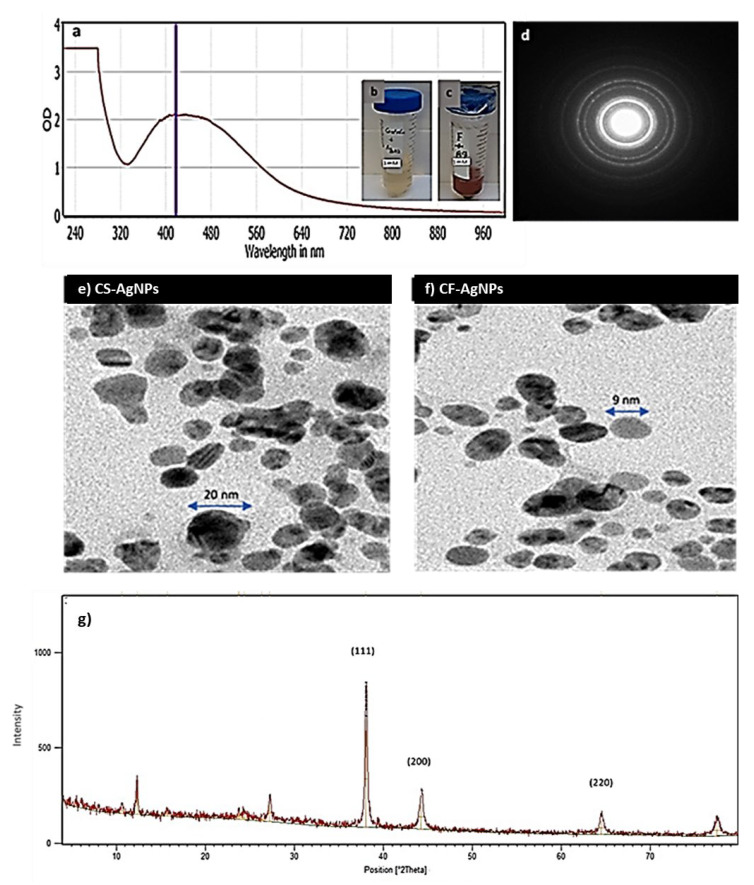
(**a**) Surface plasmon absorption bands (SPR) of AgNPs formed in the presence of *Streptomyces* sp. ASM23 cell filtrate (CF-AgNPs). (**b**) The color change due to the reduction of metal ions and (**c**) the formation of AgNPs represented by (**a**) Control+ AgNO3 and (**b**) *Streptomyces* sp. ASM23 cell filtrate+ AgNO_3_. (**d**) Selected area (electron) diffraction (SAED) confirms the crystalization of nanoparticles. (**e**) Transmission electron microscope micrographs for (**f**) AgNPs biosynthesized by *Streptomyces* sp. ASM23 cell filtrate (CF-AgNPs) and cell supernatant (CS-AgNPs). (**g**) X-ray diffraction (XRD) pattern of silver nanoparticles.

**Figure 4 microorganisms-11-00102-f004:**
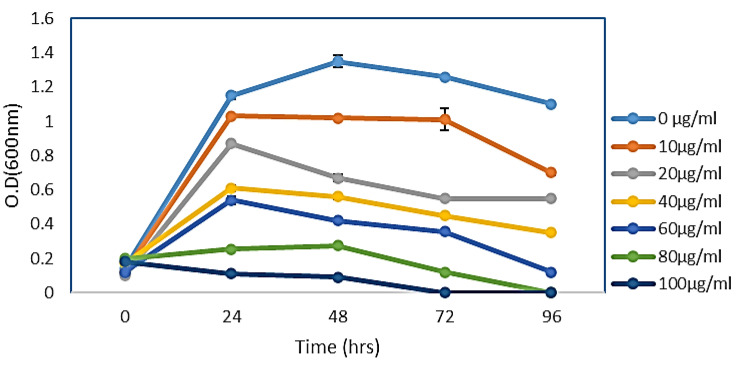
Growth curve dynamics of *E. coli* with CF-AgNPs solution. Each experiment was carried out in triplicate and the results are presented as mean values ± standard deviations (SD).

**Figure 5 microorganisms-11-00102-f005:**
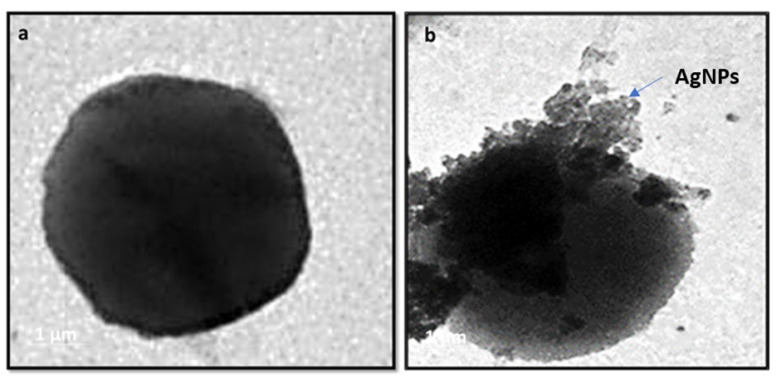
Accumulation of AgNPs on the *S. aureus* outer cell membrane (**a**) Bacteria with no treatment and (**b**) Bacteria treated with AgNPs.

**Figure 6 microorganisms-11-00102-f006:**
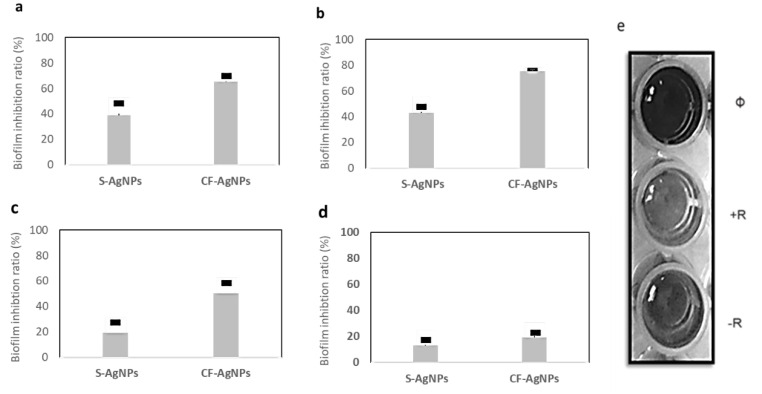
The biofilm inhibition ratio of AgNPs prepared by *Streptomyces* sp. ASM23 cell supernatant (S) and cell filtrate (CF) against four clinical microbes (**a**) *B. subtilis*, (**b**) *E. coli* (**c**) *S. aureus*, and (**d**) *P. aeruginosa*. The bars on the graph represent the mean of triplicate, and SD. (**e**) Microtiter plate, (Ø) blank, (+R) positive result, and (−R) negative result.

**Figure 7 microorganisms-11-00102-f007:**
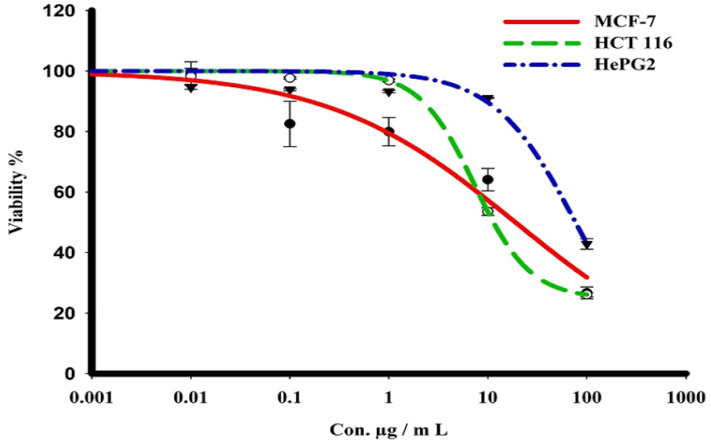
The dose–response curve of biosynthesized AgNPs from the cell supernatant of *Streptomyces* sp. ASM23 on the cytotoxicity in MCF-7, HePG2, and HCT 116 cell lines. Cells were exposed to biosynthesized AgNPs from the cell supernatant of *Streptomyces* sp. ASM23, with different concentrations of compounds for 72 h. Cell viability was detected by SRB stain.

**Figure 8 microorganisms-11-00102-f008:**
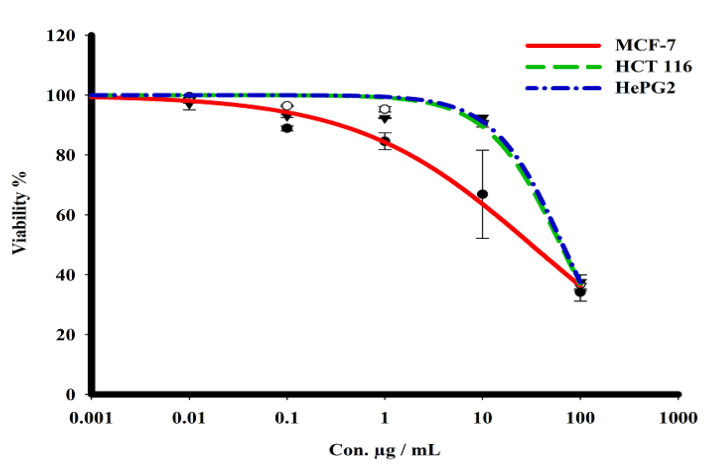
The dose–response curve of biosynthesized AgNPs from cell filtrate of *Streptomyces* sp. ASM23 on the cytotoxicity in MCF-7, HePG2, and HCT 116 cell lines. Cells were exposed to biosynthesized AgNPs from cell filtrate of *Streptomyces* sp. ASM23 with different concentrations of compounds for 72 h. Cell viability was detected by SRB stain.

**Figure 9 microorganisms-11-00102-f009:**
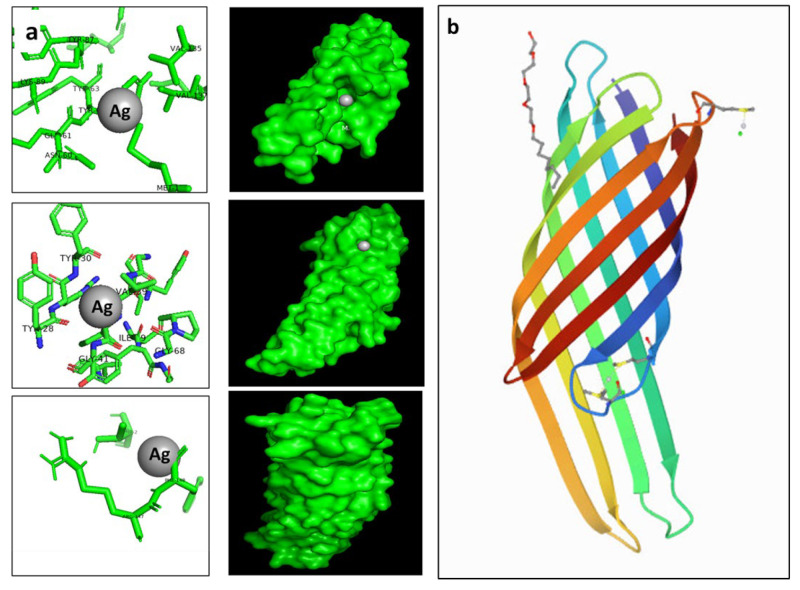
(**a**) showed the possible binding modes of AgNPs to *E. coli* outer membrane protein, while (**b**) 3D structure of *E. coli* outer membrane protein 1QJ8. Molecular docking was performed to identify the interaction of synthesized nanoparticles with amino acid residues. The docking model shows an interaction of Ag with 1QJ8 through Phe146.

**Table 1 microorganisms-11-00102-t001:** Antimicrobial activity of AgNPs produced by *Streptomyces* sp. ASM23 cell supernatant (CS-AgNPs) and cell filtrate (CF-AgNPs).

Test Microbes	Zone of Inhibition (mm)			
	S-AgNPs	CF-AgNPs	Cip	Nys
*P. aeruginosa*	11.00	14.00	18.00	-
*B. subtilis*	10.00	13.00	19.00	-
*E. coli*	15.00	19.00	18.00	-
*S. aureus*	8.00	16.00	-	
*C. albicans*	7.05	9.00	-	18
*A. niger*	-	-	-	15

All data are expressed as the mean of three reads, Cip: ciprofloxacin, Nys: Nystatin.

**Table 2 microorganisms-11-00102-t002:** The IC_50_ (µg/mL) of the biosynthesized AgNPs from the cell supernatant (CS-AgNPs) and Cell filtrate (CF-AgNPs) of *Streptomyces* sp. ASM23 against different tumor cell lines.

Sample	IC50 (μg/mL)		
	MCF-7	HepG2	HCT 116
CS-AgNPs	27.8 ± 2.3	18 ± 0.6	73.2 ± 1.5
CF-AgNPs	65.1 ± 2.4	82.9 ± 1.5	81.9 ± 1.4

## Data Availability

The data presented in this study are available on request from the corresponding author.

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
