# Peer review of "Biofilm Inhibitory Activity of Actinomycete-Synthesized AgNPs with Low Cytotoxic Effect: Experimental and In Silico Study"

_microorganisms, 2022, doi:10.3390/microorganisms11010102_

Round 1
Reviewer 1 Report
the article represent The Biofilm Inhibitory Activity of Actinomycete-Synthesized AgNPs with Low Cytotoxic Effect:as well as molecular docking insilico study
there are some minor
-abstrat: all microorganisms should be full and not abbreviative
-figure 9 need to be more clear and representative
Author Response
abstract: all microorganisms should be full and not abbreviative
Response: Thanks for your comment, done
-figure 9 need to be more clear and representative
Response: Thanks for your comment, we have modified the Figure 9 footnote to be clearer and more representative
Reviewer 2 Report
The article is well-written and examined a new actinomycete strain for its ability to bio-synthesis of an eco-friendly and less expensive silver nanoparticle (AgNPs). However, it can be further improved by considering the following comments. Please see the appendix for details.
Author Response
Thanks so much,
Reviewer 3 Report
The manuscript entitled ‘Biofilm Inhibitory Activity of Actinomycete-Synthesized AgNPs with Low Cytotoxic Effect: Experimental and in silico study’ reported a study on the antimicrobial and antibiofilm activities, as well as the cytotoxic effects of silver nanoparticles against several nosocomial pathogenic bacteria. The results are interesting. However, from the reviewer’s point of view, the manuscript needs significant improvement to be considered acceptable. Here are some of the suggestions and questions for the authors to consider and address:
1. The authors claimed that the synthesized AgNPs had certain antimicrobial and antibiofilm activities, and relatively low cytotoxicity on certain cell lines in human body, how would it be used in medical field? Is it considered safe to be consumed orally? Would there be any harmful side effects? What is the practical value of this study?
2. The English writing in this manuscript need to be improved. There are a large number of grammar errors, typos, and the use of tense is relatively messy. It is not recommended to use first person in writing a scientific journal article, instead, please use the third person. A few examples are listed: Line 31-32: results showed that the UV-vis spectrum showed... grammar mistake. particle size cannot be obtained from UV-vis spectroscopy, so it should not be written in the same sentence. The authors used perfect tense and present tense a lot, which should be past tense. I highly suggest the authors to polish the language by a native speaker before resubmission.
3. The standard deviations are missing in many figures and tables (Table 1, Figure 4,. The authors did not mention any replicate of experiments and statistical analysis in the methodology, which made the results and conclusions not that convincing. Particularly, the authors should perform formal statistical analysis (for example, ANOVA) to study the influence of AgNPs on the biofilm inhibition, antimicrobial, cytotoxic tests. These statistical analysis results will help the authors to support the results. Please add necessary information into the writing of material and methods, as well as results and discussions.
4. Definitions and use of abbreviations are messy. Please define the abbreviations in the first place they appear in the manuscript and use them as much as you can. For example, AgNPs from cell supernatant of Streptomyces spp ASM23, and AgNPs from cell filtrate of Streptomyces spp ASM23 showed up everywhere in the text, while S-AgNP, CF-AgNP and AgNP (supernatant), AgNP (cell filtrate) have been used mix and match in figures and tables, which made it very confusing.
5. Please add the detailed information on the chemicals and reagents used in the study, including producer, town/city. province/state and country. Similarly, detailed information should be added to the equipment and measuring device used,.
6. Section 3.4: Line 256-263 belong to the material and methods section and seem to be repetitive to the introduction in section 2.6
7. Please re-edit Figure 3. The numbers of the subplots are hard to recognize. The wordings in the figures are blurry. Why did the authors used Figure 3j after 3f? where were 3g, 3h, and 3i???
8. Line 284-289: redundant, just the same thing repeated 4 times, keep the writing concise.
9. Figure 5: could the authors mark the AgNPs in the figure 5 to support their statement on the accumulation of AgNPs on cell membrane, as well as AgNPs inside the cell. Particularly, which part is the AgNP? Currently, it seems like only the breaking of cell membrane can be observed.
10. Line 399-416: These two paragraphs seem to be introductions, not discussions of the obtained results from this study.
11. Please add numerical values in the conclusion to highlight and support the significant findings and values of this study.
Author Response
Response to Reviewer 3 Comments
- The authors claimed that the synthesized AgNPs had certain antimicrobial and antibiofilm activities, and relatively low cytotoxicity on certain cell lines in human body, how would it be used in medical field? Is it considered safe to be consumed orally? Would there be any harmful side effects? What is the practical value of this study?
Response: Thanks for your comment, ongoing investigations and extensive studies evidenced the potential use of AgNPs in vaccine and drug carriers for specific and selective cell or tissue targeting. Additionally, silver nanoparticles have recently been used as a sterilizing nanomaterial in medical textile products through the functionalization of fibers and fabrics achieving textiles with antibacterial and antifungal properties to prevent hospital nosocomial infection. It also could be used in catheter Modification, dental Applications, and bone Healing and other medical applications.
- The English writing in this manuscript need to be improved. There are a large number of grammar errors, typos, and the use of tense is relatively messy. It is not recommended to use first person in writing a scientific journal article, instead, please use the third person. A few examples are listed: Line 31-32: results showed that the UV-vis spectrum showed... grammar mistake. particle size cannot be obtained from UV-vis spectroscopy, so it should not be written in the same sentence. The authors used perfect tense and present tense a lot, which should be past tense. I highly suggest the authors to polish the language by a native speaker before resubmission.
Response: Thanks for your comment, we have sent the article to the specialized office for proofreading and grammar check. all changes have been highlighted
- The standard deviations are missing in many figures and tables (Table 1, Figure 4,. The authors did not mention any replicate of experiments and statistical analysis in the methodology, which made the results and conclusions not that convincing. Particularly, the authors should perform formal statistical analysis (for example, ANOVA) to study the influence of AgNPs on the biofilm inhibition, antimicrobial, cytotoxic tests. These statistical analysis results will help the authors to support the results. Please add necessary information into the writing of material and methods, as well as results and discussions.
Response: Thanks for your comment, done
- Definitions and use of abbreviations are messy. Please define the abbreviations in the first place they appear in the manuscript and use them as much as you can. For example, AgNPs from cell supernatant of Streptomyces spp ASM23, and AgNPs from cell filtrate of Streptomyces spp ASM23 showed up everywhere in the text, while S-AgNP, CF-AgNP and AgNP (supernatant), AgNP (cell filtrate) have been used mix and match in figures and tables, which made it very confusing.
Response: Thanks for your comment, we have defined the abbreviation in the first place and changed all others accordingly
- Please add detailed information on the chemicals and reagents used in the study, including producer, town/city. province/state and country. Similarly, detailed information should be added to the equipment and measuring device used,.
Response: Thanks for your comment, done
- Section 3.4: Line 256-263 belong to the material and methods section and seem to be repetitive to the introduction in section 2.6
Response: Thanks for your comment, we have deleted it.
- Please re-edit Figure 3. The numbers of the subplots are hard to recognize. The wordings in the figures are blurry. Why did the authors used Figure 3j after 3f? where were 3g, 3h, and 3i???
Response: Thanks for your comment, done
- Line 284-289: redundant, just the same thing repeated 4 times, keep the writing conci
Response: Thanks for your comment, we have deleted the repeated sentence
- Figure 5: could the authors mark the AgNPs in the figure 5 to support their statement on the accumulation of AgNPs on cell membrane, as well as AgNPs inside the cell. Particularly, which part is the AgNP? Currently, it seems like only the breaking of cell membrane can be observed.
Thanks for your comment, done
- Line 399-416: These two paragraphs seem to be introductions, not discussions of the obtained results from this study.
Thanks for your comment, done
- Please add numerical values in the conclusion to highlight and support the significant findings and values of this study.
Thanks for your comment, we have done this
Reviewer 4 Report
Reviewer's comments on increasing the scientific value of the publication:
- It is a pity that the Aspergillus fumigatus fungus strain was not used in the research
- Were the soil samples used in the study previously tested for actinomycetes?
- To which antimicrobials have the bacterial strains obtained from the hospital in Egypt been shown to be resistant?
Author Response
Reviewer's comments on increasing the scientific value of the publication:
- It is a pity that the Aspergillus fumigatus fungus strain was not used in
the research
Response: Thanks for your comment
- Were the soil samples used in the study previously tested for actinomycetes?
Response: Thanks for your comment, we are not quite sure if the soil from the location have studied before or not? But our information that no one study the microbes of this soil before for biosynthesis of Nanometals
- To which antimicrobials have the bacterial strains obtained from the
hospital in Egypt been shown to be resistant?
Response: Thanks so much for your comment, the obtained microbes were tested at al Azhar university hospital toward several antibiotic. It was isolated from patients with bacterial infections. Name of the hospital
Round 2
Reviewer 3 Report
The reviewer has suggested the authors to perform formal statistical analysis on their experimental results to support their justification, and the authors responded that they have done it. But in the revised manuscript, it is not seen. Adding only the standard deviation values is not formal statistical analysis! Variability of microbial test results is usually relatively large, without statistical analysis results the conclusions cannot be made. The reviewer insists that formal analysis must be added, for example the ANOVA analysis with mean comparison tests.
Author Response
Thanks so much for your comment
we have done statistical analysis to the data for the data using Anova to find if the results are statistically significant or not.